# Bird's-eye-view Informed Reasoning Driver

**Yinuo Wang**[1][†], **Mining Tan**[3,4][†], **Yuanxin Zhong**[2][†][*], **Zhitao Wang**[2], **Siyuan Cheng**[2][*]

[1] Tsinghua University, [2] Huawei Inc., [3] University of Chinese Academy of Sciences

[3] MAIS, Institute of Automation, Chinese Academy of Sciences

cmpute@gmail.com, tanmining24@mails.ucas.ac.cn

## Abstract

Motion planning in complex environments remains a core challenge for autonomous driving. While existing rule-based or imitation learning-based motion planning methods perform well in common scenarios, they often struggle with complex, long-tail scenarios. To address this problem, we introduce the Bird's-eye-view Informed Reasoning Driver (BIRDriver), a hierarchical framework that combines a Vision-Language Model (VLM) with a motion planner. BIRDriver leverages the commonsense reasoning capabilities of the VLM to effectively handle these challenging long-tail scenarios. Unlike prior methods that require domain-specific encoders and costly alignment, our approach compresses the environment into a single-frame bird's-eye-view (BEV) map, a paradigm that enables the model to fully leverage its knowledge from internet-scale pre-training. It then generates high-level key points, which are encoded and passed to the motion planner to produce the final trajectory. However, a major challenge is that standard VLMs struggle to generate the precise numerical coordinates required for such key points. We address this limitation by fine-tuning them on a composite dataset of three auxiliary types to enhance spatial localization, scene understanding, and key-point generation, complemented by a token-level weighted mechanism for improved numerical precision. Experiments on the nuPlan dataset demonstrate that BIRDriver outperforms the base motion planner in most cases on both Test14-hard and Test14-random benchmarks, and achieves state-of-the-art (SOTA) performance on the InterPlan long-tail benchmark.

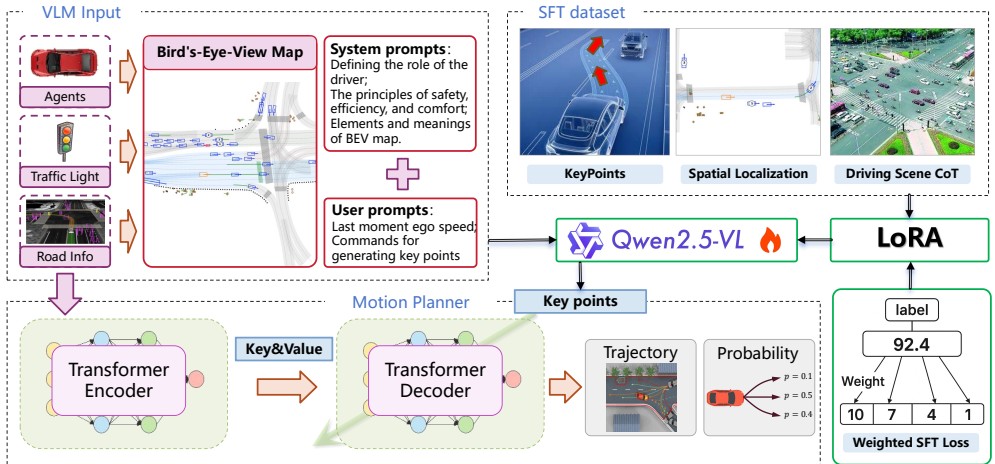

Figure 1: **The architectural overview of BIRDriver.** BIRDriver consists of two main components: a VLM and a motion planner. The input to the VLM includes a single-frame BEV map, a system prompt, and a user prompt. The VLM then generates key points, which are subsequently encoded and fed into the decoder module of the motion planner. The base motion planner generates a concrete trajectory based on the key points and structured feature information.

---

*Corresponding author

# 1 INTRODUCTION

Motion planning (Guan et al., 2022; Duan et al., 2024; Zhu et al., 2026; Zou et al., 2025) is a fundamental component of autonomous driving, utilizing outputs from perception modules to generate future poses for the ego vehicle. The performance of the motion planner plays a crucial role in ensuring vehicle safety, operational efficiency, and successful navigation through complex environments. Recent advances in deep learning (Huang et al., 2023; Cheng et al., 2024a; Zheng et al., 2025) have led to notable progress in trajectory planning. However, due to the complexity and uncertainty inherent in open-world autonomous driving, current methods based on imitation learning struggle to handle long-tail scenarios. As shown in Fig. 3, existing state-of-the-art (SOTA) methods struggle to handle the long-tail scenario of overtaking a stalled vehicle via a detour, primarily because such cases are absent from the training data.

Recently, LLMs (Achiam et al., 2023; Liu et al., 2024a; Yang et al., 2025) and VLMs (Chen et al., 2024c; Hurst et al., 2024; Bai et al., 2025) have attracted increasing attention. Leveraging inherent reasoning capabilities, these models exhibit stronger generalization to zero-shot and few-shot scenarios, as demonstrated in various domains (Liu et al., 2024b; Saha et al., 2024; Chen et al., 2025; Wang et al., 2026; Li et al., 2026). Consequently, this makes them highly promising for assisting motion planners, particularly in addressing complex long-tail scenarios. In terms of how to incorporate the intentions provided by VLMs or LLMs into the base motion planner, existing approaches can be categorized into three types: meta-actions, hidden features of LLMs or VLMs, and waypoints. In the first type, Senna (Jiang et al., 2024) discretizes high-level driving intentions into a fixed set of meta-actions, which are encoded and injected into the base end-to-end driving model. However, the granularity of such intention-transmission methods in decision making remains limited. In the second type, AsyncDriver (Chen et al., 2024b) introduces an asynchronous framework that takes advantage of the hidden states of the final layer of LLM to guide motion planning. However, hidden features are abstract and often lack interpretability, making it difficult to trace the underlying high-level intentions. In the third type, DriveVLM (Tian et al., 2024) directly generates a sequence of waypoints, which are subsequently refined by a base end-to-end planner. However, trajectory generation offers limited benefits from LLM pretraining, as the required capabilities mainly stem from domain-specific driving data rather than internet-scale corpora. Furthermore, high-level driving intentions can often be conveyed without long waypoint sequences, while the reliance on dozens of waypoints adds redundancy and increases computational complexity.

Unlike existing approaches, in this work, we propose using no more than three key points to convey high-level driving intentions. Each key point represents information relative to the ego vehicle. Except for the final key point, which represents the end position of the trajectory, the remaining key points are time-agnostic and primarily capture geometric features of the trajectory. In addition, compared to existing VLM-based approaches, our method uses only a single-frame BEV map as input for the VLM to understand the current autonomous driving scene. No scene-specific information is included in the text inputs.

The contributions of this paper are the following:

- We propose Bird's-eye-view Informed Reasoning Driver (BIRDriver), the first hierarchical VLM–motion planning framework that uses a BEV map as its primary input. Our method conveys all information about the current driving scene solely with a BEV map, with no scene-specific details provided in the text inputs. By operating on the BEV representation, BIRDriver avoids the need to align heterogeneous sensor data across different vehicle platforms, thereby simplifying deployment in diverse real-world settings.

- We introduce the use of relative $(x, y, \phi)$ key points, rather than complete trajectories, to convey high-level driving intentions. Furthermore, to enable the VLM to generate these key points accurately, we develop a fine-tuning strategy that leverages three specialized datasets and incorporates a weighted supervised fine-tuning (SFT) loss.

- Extensive experiments on the nuPlan dataset demonstrate that our method achieves SOTA performance in InterPlan, a long-tail test scenario. In both the Test14-hard and Test14-random sets, it outperforms the base motion planner in most cases. Specifically, in Inter-Plan, our method outperforms PLUTO (Cheng et al., 2024a) and Diffusion Planner (Zheng et al., 2025) by **13.0%** and **38.8%**, respectively.

## 2 RELATED WORKS

### 2.1 MOTION PLANNING FOR AUTONOMOUS DRIVING

The motion planner is a core module in autonomous driving systems, responsible for generating a safe and comfortable trajectory based on structured environmental information. While traditional planners rely on hand-crafted rules, the field has increasingly shifted to learning-based methods that learn driving policy directly from data (Chen et al., 2024a). These modern planners have achieved state-of-the-art performance on challenging benchmarks (Jiang et al., 2023; Huang et al., 2023; Cheng et al., 2024b;a). However, a fundamental limitation persists: these planners make decisions based on structured inputs like object states and map elements, but lack a deeper, human-like contextual understanding. They struggle to interpret ambiguous situations or handle scenarios that require commonsense knowledge beyond the scope of their training data, creating a need for models with broader reasoning capabilities.

### 2.2 VLMS FOR AUTONOMOUS DRIVING

The commonsense reasoning and generalization capabilities of VLMs hold promise for enhancing autonomous driving. Existing research typically adopts one of two architectures. The first uses VLMs as end-to-end models to directly produce trajectories or control signals along with interpretable justifications (Mao et al., 2023; Shao et al., 2024; Xu et al., 2024; Yuan et al., 2024), but such use raises safety concerns in high-frequency planning. As a result, hierarchical frameworks are gaining traction. In these, a real-time motion planner handles trajectory generation, while the VLM plays a supervisory role. Some studies use the VLM for high-level strategic guidance (Jiang et al., 2024; Long et al., 2024), while others focus on safety and interpretability, with the VLM offering corrective feedback or intermediate semantic variables (Ding et al., 2024; Chen et al., 2024b; Qian et al., 2025). Following this paradigm, we employ the VLM to generate key points that guide a downstream motion planner. However, a key challenge in VLM-based systems is their limited spatial awareness in complex visual scenes (Jiang et al., 2025).

### 2.3 GROUNDING VLM WITH BEV REPRESENTATIONS

Addressing the limited spatial awareness of VLM requires grounding its reasoning in a structured spatial representation. The BEV format is well-suited for this purpose, offering a unified top-down view ideal for planning tasks (Philion & Fidler, 2020). Recent works have begun using BEV to enhance VLM capabilities. One direction grounds situational understanding in BEV to support language-based scene queries (Choudhary et al., 2024; Xu et al., 2025), while another uses BEV as a direct context for high-level planning, allowing the VLM to produce strategic maneuvers or semantic goals (Zheng et al., 2024; Winter et al., 2025). Although these efforts highlight the value of BEV, they often rely on additional modalities or focus on querying. In contrast, we propose a planning framework where the reasoning of VLM is grounded solely in the BEV representation.

## 3 PRELIMINARY

**Ramer–Douglas–Peucker (RDP)** (Ramer, 1972) algorithm is a classic method for curve simplification and key point extraction. It recursively identifies representative points that preserve the essential shape of a trajectory, thereby achieving data compression within a specified error tolerance. Given a curve defined by a set of points $\{P_i\}_{i=1}^N$, the RDP algorithm initially considers the straight line segment that connects the end points $P_1$ and $P_N$. It computes the perpendicular distance from each intermediate point $P_i$ $(1 < i < N)$ to this line as follows:

$$d_i = \frac{|(P_N - P_1) \times (P_i - P_1)|}{\|P_N - P_1\|}. \tag{1}$$

The point with the maximum distance is then identified:

$$d_{\max} = \max_{1 < i < N}(d_i). \tag{2}$$

If $d_{\max}$ is greater than a preset tolerance $\epsilon$, then this point is split into two parts and the algorithm is recursively called on both parts. Otherwise, only endpoints are kept.

**Supervised Fine-Tuning (SFT) loss** (Achiam et al., 2023) is a conditional language modeling loss that is basically an autoregressive cross-entropy loss. It is fed with an input-output pair $(X, Y)$, and maximizing conditional probability $p_\theta(Y|X)$ yields the following objective:

$$\mathcal{L}_{\text{SFT}}(\theta) = -\mathbb{E}_{(X,Y)\sim D}\left[\sum_{t=1}^{T} \log p_\theta(y_t|y_{<t}, X)\right], \tag{3}$$

where $D$ denotes the manually curated supervised dataset and $T$ is the length of the output sequence.

## 4 METHODS

In this section, we detail the design of our proposed BEV-Informed Reasoning Driver (BIRDriver). Fig. 1 illustrates the overall architecture of BIRDriver. Our method employs a hierarchical architecture comprising a VLM and a motion planner, which achieves closed-loop autonomous driving through decoupled training and sequential inference. The input to the VLM includes a single BEV map, a system prompt, and a user prompt, without requiring multi-frame surround-view camera images. The VLM generates key points, which are subsequently encoded into high-dimensional feature representations via the KeyPoint Encoder. Finally, the motion planner generates future trajectories by taking the structured information of the traffic scene and the key points as input. Specific details of the system and user prompts are provided in Appendix A.

The remainder of this section is organized as follows. We begin by describing the construction of the BEV map and the key points used to represent the driving scene. We then detail the dataset creation process and the parameter-efficient fine-tuning strategy applied to the VLM for key point generation. Subsequently, we present our methods for enhancing key point prediction accuracy, which involve the design of task-specific datasets and the adoption of a weighted loss function. Finally, we outline the fine-tuning procedure of the motion planner.

### 4.1 DRIVING SCENE REPRESENTATION

Driving scene representation is a crucial component of hierarchical autonomous driving. We pioneer a hierarchical framework that, unlike camera-based approaches, relies solely on a single-frame BEV map as the visual input for the VLM. This design mitigates inconsistencies across camera types and simplifies data collection and utilization (Ho et al., 2024). In addition, to ensure the VLM can efficiently interpret the BEV map, we explicitly explain the symbolic representations of different driving elements in the system prompt.

We extract five categories of information from the environment: map, agent, traffic lights, route, and obstacles. Map information includes lanes, lane connectors, crosswalks, and discrete waypoints. Agent information includes the ego vehicle, other vehicles, bicycles, and pedestrians, which are represented by orange, blue, pink, and brown bounding boxes, respectively. An arrow is added to each vehicle's bounding box to indicate its driving direction. Notably, the trajectories of all non-ego agents are depicted using green solid lines to show their movements over the past two seconds. Traffic light status of the ego lane is encoded as the color of the corresponding stop line at the intersection. Route information includes two components: the drivable area, which is filled with light blue, and the reference lines, which are indicated by purple arrows. Obstacle information includes three types of objects: construction signs, roadblocks, and traffic cones, which are represented by black bounding boxes. These five types of information are rendered together to generate the BEV map shown in Fig. 2.

### 4.2 KEY POINT EXTRACTION

The future trajectory of the ego vehicle is represented as a sequence of poses $(x_i, y_i, \phi_i)_{i=1}^{N}$, where $x_i$, $y_i$, and $\phi_i$ denote the longitudinal position, lateral position, and heading angle in the frame of ego vehicle, respectively. Instead of directly using this dense temporal sequence, we extract a sparse set of key points that compactly encode the underlying driving intention.

Specifically, we apply the RDP algorithm to select representative key points. Since different types of maneuvers (e.g., lane keeping, lane changing, turning) exhibit varying levels of complexity, we

adaptively adjust the maximum allowable number of key points according to the trajectory type. In all cases, the final point of the trajectory is always retained as a key point.

### 4.3 DATASET DESIGN AND VLM FINE-TUNING

Although system prompts provide explanations for the elements of BEV map, existing VLMs still exhibit a limited capacity to generate effective key points directly from this representation. Therefore, we employ LoRA-based (Hu et al., 2022; Dettmers et al., 2023; Hayou et al., 2024) parameter-efficient fine-tuning (PEFT) and construct a specialized dataset to enable the VLM to generate meaningful key points. This fundamental dataset, named the *Key Point Dataset*, takes as input a BEV map, a system prompt, and a user prompt. The system prompt describes the meanings of the various elements in the BEV map, relevant traffic rules, and safe driving requirements. The user prompt adopts diverse question formats to guide the VLM in generating key points. The output of VLM consists of key points represented in textual form.

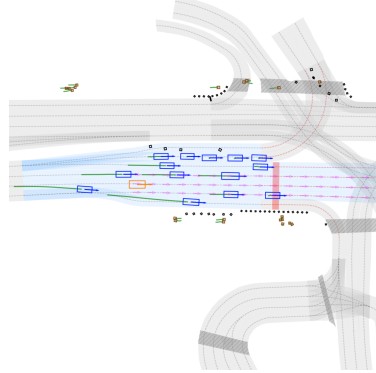

Figure 2: **BEV map.** The orange box denotes the ego vehicle, while the blue boxes represent surrounding vehicles. Arrows indicate the direction of vehicle movement. The green solid line illustrates the historical trajectory of the past 2 seconds.

However, initial experiments revealed that the key point prediction accuracy on the test set remained relatively low. Such large errors not only fail to improve the performance of autonomous driving systems in long-tail scenarios, but may even degrade the quality of the generated trajectories and lead to lower scores. To address this core issue, we propose methods from two perspectives: designing a diversified, multi-task dataset and developing a specialized SFT loss function.

#### 4.3.1 DATASET DESIGN.

The accuracy of key point prediction is mainly limited by two factors: the VLM's insufficient understanding of the correspondence between distances in the BEV pixel space and real-world physical distances, and its limited ability to classify and interpret driving scenes. To address these issues, we first construct the *Spatial Localization Dataset*, where the VLM predicts the poses of randomly selected vehicles relative to the ego vehicle, thereby bridging the gap between pixel and physical distances. We then design the *Driving Scene Stepwise Dataset*, which requires the VLM to identify the driving scene type prior to predicting key points, enhancing its scene comprehension. Finally, we perform supervised fine-tuning on a composite dataset primarily based on the *Key Point Dataset*, augmented with the other two datasets in specified proportions.

#### 4.3.2 WEIGHTED SFT LOSS FUNCTION.

In a typical SFT loss, each token is treated equally. However, in the key points prediction task, the loss associated with numeric tokens plays a more important role. To address this issue, we assign higher loss weights to three categories of tokens: numeric tokens, decimal points, and sign symbols. Considering that higher-order digits are more critical for precision than lower-order digits, we design a tiered weighting scheme in which the loss weights decay linearly from the most significant token to the least significant within a number, and the sign tokens are assigned the highest loss weight by default. It is worth noting that all floating-point labels are rounded to two decimal places. The specific formulation can be expressed as follows:

$$\mathcal{L}_{\text{SFT}}(\theta) = \frac{1}{T} \sum_{t=1}^{T} w_t \left[ -\log p_\theta(y_t \mid y_{<t}, X) \right],$$

$$w_t = \begin{cases} (\alpha + d_n) - k\dfrac{(\alpha + d_n) - 1}{L_n - 1}, & t = t_k^{(n)}, \ L_n > 1, \\ \alpha + d_n, & t = t_0^{(n)}, \ L_n = 1, \\ 1, & t \notin \bigcup_{n \in \mathcal{N}} \{t_k^{(n)}\}, \end{cases} \quad (4)$$

where $T$ denotes the shifted target length; $\alpha > 0$ is a hyperparameter (set to 5 in this paper); $y_t$ is the token at position $t$; $p_\theta(y_t \mid y_{<t}, X)$ represents the model's conditional probability; $\mathcal{N}$ is the set of contiguous number segments matched by a specific pattern; $d_n$ denotes the number of digits in segment $n$; $L_n$ is its total token length; $t_k^{(n)} = t_0^{(n)} + k$ indicates the index of the $k$-th token in segment $n$; and $w_t$ decays linearly from $(\alpha + d_n)$ at the first token of a segment to 1 at its last token, while tokens outside any number segment are assigned $w_t = 1$.

Compared to the Position-Dependent Cross-Entropy (PDCE) loss (Zhang et al., 2025), our approach is considerably simpler to implement, as it avoids the complexities of generating soft targets and standardizing numeric formats. More importantly, by only adjusting loss weights rather than altering the target distribution itself, our method mitigates the risk of impairing the model's general language capabilities, a potential side effect of approaches like PDCE.

## 4.4 FINE-TUNING MOTION PLANNER

The goal of fine-tuning the motion planner is to ensure it can accurately follow the provided key points. We use the PointEncoder module from PLUTO (Cheng et al., 2024a) to encode key point information and integrate it with the original query features. We choose PLUTO as our base motion planner because, under identical encoder and fine-tuning settings, Diffusion Planner (Zheng et al., 2025) demonstrates weaker performance in tracking multiple key points.

This fine-tuning process is conducted independently of the VLM. We use the key point extraction function proposed in the previous section to extract key points from real trajectories. To enhance robustness, we augment the training data by adding Gaussian noise with zero mean and a standard deviation equal to the mean absolute error of the key points predicted by the VLM to the extracted ground-truth key points. Specifically, it can be formulated as:

$$(\tilde{x}_i, \tilde{y}_i, \tilde{\phi}_i) = (x_i, y_i, \phi_i) + \epsilon_i, \quad \epsilon_i \sim \mathcal{N}\Big(0, \Sigma\Big),$$
$$\Sigma = \mathrm{diag}(\sigma_x^2, \sigma_y^2, \sigma_\phi^2), \tag{5}$$

where $\sigma_x$, $\sigma_y$, and $\sigma_\phi$ denote the mean absolute errors of the VLM predicted key points in the dimensions $x$, $y$, and $\phi$, respectively, and $i$ represents the $i$-th key point. It should be noted that in the joint inference stage, we no longer add noise to the key points predicted by the VLM. Furthermore, to enhance temporal consistency in decision-making during the inference phase, we use the final planned point from the previous timestep as an additional key point for the current timestep, feeding it to the planner alongside the key points provided by the VLM.

## 5 EXPERIMENTS

### 5.1 EXPERIMENTAL SETUPS

**Baselines.** We categorize existing methods into four groups based on their trajectory generation approach: Rule-based, Imitation Learning (IL), Large Language Models (LLM), and Vision-Language Model-Imitation Learning (VLM-IL). The first three groups include the following representative algorithms, while our method belongs to the fourth category. (1) PDM (Dauner et al., 2023), the nuPlan Challenge winner, includes a neural network variant (PDM-Open) and a rule-based version (PDM-Closed). (2) UrbanDriver (Scheel et al., 2022) employs policy gradient optimization. (3) GameFormer (Huang et al., 2023) integrates level-$k$ reasoning with a Transformer for multi-agent planning. (4) PlanTF (Cheng et al., 2024b) mitigates closed-loop error through perturbation normalization and future correction. (5) PLUTO (Cheng et al., 2024a) improves robustness via perceptual augmentation and contrastive learning. (6) Diffusion Planner (Zheng et al., 2025) achieves state-of-the-art performance using diffusion models and classifier guidance. (7) Instruct-Driver (Zhang et al., 2024) fine-tunes an LLM for instruction-driven planning. (8) PlanAgent (Zheng et al., 2024) integrates BEV, lane semantics, and chain-of-thought (CoT) reasoning with a VLM for robust planning.

**Benchmarks and Metrics.** We evaluate our methods using the Test14-random, Test14-hard (Cheng et al., 2024b), and InterPlan (Hallgarten et al., 2024) benchmarks. Specifically, Test14-random consists of 261 scenarios, Test14-hard includes 272 challenging scenarios, and InterPlan

represents a long-tail scenario simulation dataset. Evaluation is based on the closed-loop score (CLS) provided by the official nuPlan devkit in both non-reactive (CLS-NR) and reactive (CLS-R) closed-loop settings. The final score is calculated as the average across all scenarios, ranging from 0 to 100, with higher scores indicating better algorithm performance.

**Implementation Details.** We use the nuPlan platform, a large-scale closed-loop simulator for autonomous driving trajectory planning. It includes over 1,500 hours of expert driving data from four cities, and supports complex scenarios such as intersections, roundabouts, and pedestrian interactions. The simulator replays real-world scenarios with non-ego agents controlled by either log replay or an Intelligent Driver Model policy, while the ego vehicle executes user-defined trajectories. When using the RDP algorithm to generate keypoints, we set the parameter $\epsilon$ to 0.02. For parameter-efficient fine-tuning of the VLM, we adopted LoRA technique, applying adapters to all linear layers. The language model was unfrozen during training. Training was performed on eight NVIDIA H800 GPUs for five epochs, using AdamW with cosine annealing. The dataset consists of 838,824 samples, with a category ratio of 10:1:2 for *Key Point*, *Spatial Localization*, and *Driving Scene Stepwise* tasks. For motion planner fine-tuning, we followed the PLUTO framework to create 1 million data splits. Training was conducted on eight RTX 4090 GPUs with a batch size of 128 for 10 epochs. For more details, refer to Appendix B.

## 5.2 MAIN RESULTS

| Type | Method | Test14-random | | Test14-hard | | InterPlan |
|---|---|---|---|---|---|---|
| | | CLS-NR | CLS-R | CLS-NR | CLS-R | CLS-R |
| Rule-based | PDM-Closed | 90.05 | 91.64 | 65.07 | 75.18 | 43.51 |
| IL-based | PDM-Open | 52.83 | 57.22 | 33.50 | 35.85 | 22.83 |
| | UrbanDriver | 63.88 | 61.01 | 49.43 | 49.67 | 8.42 |
| | GameFormer | 80.91 | 79.93 | 69.98 | 66.82 | 14.29 |
| | PlanTF | 85.98 | 80.53 | 70.10 | 61.49 | 36.53 |
| | PLUTO | 91.87 | 90.03 | 80.03 | 76.92 | 48.92 |
| | Diffusion Planner | **93.85** | **91.73** | 78.82 | **81.42** | 39.85 |
| LLM&VLM-based | InstructDriver (LLM-based) | 70.31 | 66.96 | 57.37 | 52.95 | 32.31 |
| | PlanAgent (VLM-based) | - | - | 72.51 | 76.82 | - |
| VLM&IL-based | BIRDriver (PLUTO) | 91.46 | 91.26* | **80.56*** | 80.33* | **55.29*** |

Table 1: Comparison of various planners across the Test14-random, Test14-hard, and InterPlan datasets. An asterisk (*) indicates performance exceeding that of the baseline motion planner, PLUTO. BIRDriver demonstrates a substantial advantage on the long-tailed InterPlan dataset. The VLM base model employed is Qwen2.5VL-3B.

We compare our method, BIRDriver, against a wide range of existing planners on the nuPlan benchmark, as shown in Table 1. Except for the CLS-NR results on Test14-random, BIRDriver achieves better performance than PLUTO across all three benchmarks, where PLUTO serves as the baseline algorithm for our motion planner. Specifically, on the Test14-hard benchmark, the scores for CLS-NR and CLS-R improve by 0.53 and 3.41 points, respectively. This indicates that our dual-system autonomous driving framework, VLM-Planner, effectively enhances the performance of the base motion planner. On the most challenging InterPlan benchmark, which features numerous zero-shot scenarios, such as driving through construction zones and overtaking stalled vehicles, our method achieves the best performance, surpassing PLUTO and Diffusion Planner by 13.0% and 38.8%, respectively. We attribute this significant improvement to the generalist capabilities of the VLM. Taking the lane-changing overtaking scenario as an example, the ego vehicle must wait for oncoming traffic to pass before temporarily entering the opposite lane to overtake a stalled vehicle. Existing state-of-the-art methods fail to manage such situations. In contrast, our approach enables the VLM to first generate a key point for following the lead vehicle, and after the oncoming vehicle passes, to generate a feasible overtaking key point. Results are shown in Fig. 3.

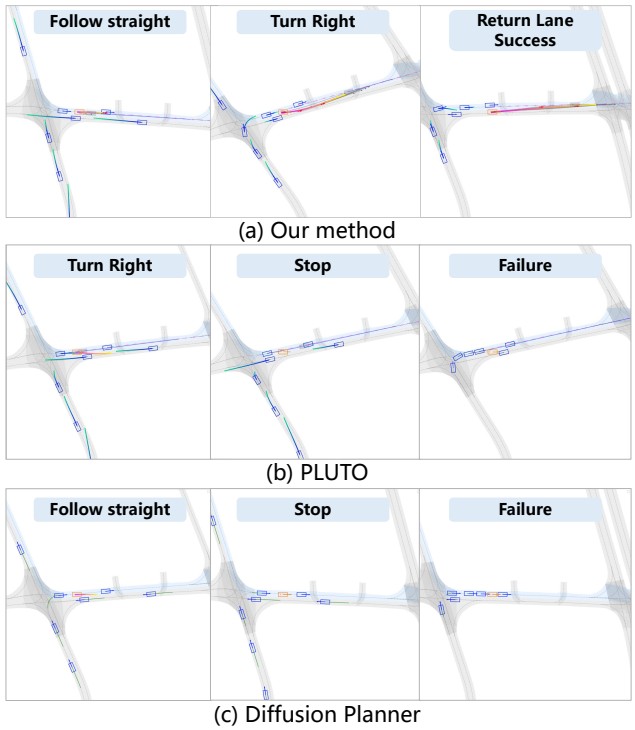

Figure 3: **Performance in the lane-changing overtaking scenario.** Red arrows denote the predicted key points. Unlike existing SOTA methods, only our method successfully completes this scenario.

## 5.3 ABLATION AND ANALYSIS

In this section, we conduct an ablation study to evaluate the effects of dataset design and weighted SFT loss on key point prediction accuracy. Additionally, we investigate how different key point prediction methods and the choice of base VLMs during fine-tuning influence overall performance.

**Impact of dataset design variations.** To evaluate the effectiveness of different dataset designs, we first construct a test dataset comprising 272 clips from Test14-hard. We then use Qwen2.5VL-3B (Bai et al., 2025) as the base VLM for evaluation and apply our weighted SFT loss. The ego-centric coordinate system is defined such that the $x$-axis aligns with the vehicle's forward direction, while the $y$-axis is perpendicular to it, pointing to the left side of the vehicle. The angle $\phi$ represents the orientation, where a left turn is considered positive. Table 2 presents the mean absolute values of the relative errors.

The results demonstrate that both the *Driving Scene Stepwise* and the *Spatial Localization* dataset effectively reduce key point prediction errors. In particular, the latter achieves more substantial improvement, suggesting that reducing the discrepancy between distances in the BEV map and those in the physical world plays a more pivotal role in minimizing prediction errors. Finally, compared to using only the *KeyPoint* dataset, the prediction errors for $x$, $y$, and $\phi$ are reduced by 11.9%, 20.0%, and 10.2%, respectively.

**Effect of Weighted SFT Loss.** A comparison between the last and second-to-last rows of Table 2 demonstrates that the weighted SFT loss substantially reduces prediction errors. Specifically, the errors in predicting $x$, $y$, and $\phi$ decrease by 10.9%, 9.2%, and 8.0%, respectively.

**Influence of key point extraction methods.** In this paper, we adopt the RDP algorithm as the core method for generating key points. In this subsection, we consider only the final point of each trajectory as the key point. The model is fine-tuned using InternVL2.5-4B (Chen et al., 2024c)

| Dataset Type | $x$-error | $y$-error | $\phi$-error |
|---|---|---|---|
| *KeyPoint* | 4.27m | 1.35m | 4.23° |
| *+Driving Scene Stepwise* | 4.17m | 1.28m | 4.20° |
| *++Spatial Localization* | **3.76m** | **1.08m** | **3.80°** |
| Without weighted SFT loss | 4.22m | 1.19m | 4.13° |

Table 2: Mean absolute relative errors in key point predictions by Qwen2.5VL-3B after fine-tuning on different dataset combinations. The last row represents the setting using the full dataset without the weighted SFT loss.

and evaluated on the InterPlan dataset. The results, presented in Table 3, demonstrate that the key points generated by our method provide more effective guidance to the motion planner compared to using only the final trajectory point in long-tail scenarios. Moreover, we find that using only the final waypoint as the key point leads to worse performance than the base motion planner, PLUTO. This degradation is likely caused by the absence of guidance from intermediate key points, which prevents the base motion planner from effectively generating a trajectory that reaches the sole key point, ultimately resulting in planning failure.

| Key point extraction methods | InterPlan |
|---|---|
| RDP (Our method) | 53.81 |
| Final trajectory point | 34.72 |

Table 3: Performance of the InternVL2.5-4B model trained with different key point extraction methods on the InterPlan dataset.

**Effect of base VLM selection.** To investigate the impact of the size of the VLM parameter on the precision of the prediction, we performed experiments using InternVL2.5-2B and 4B, as well as Qwen2.5VL-3B and 7B, under the same data set and settings of the weighted loss function. The results are presented in Table 4. Compared to InternVL2.5-2B, InternVL2.5-4B reduces the prediction errors for $x$, $y$, and $\phi$ by 43.7%, 48.4%, and 56.3%, respectively. This substantial gap indicates that when the model has too few parameters, the quality of the generated key points is significantly lower than that produced by larger models. In addition, compared to Qwen2.5VL-3B, Qwen2.5VL-7B reduces the relative error in the $x$ direction by 4.3%, while the errors in the $y$ direction and $\phi$ remain nearly unchanged or slightly increase. This suggests that once the parameter size reaches a certain threshold, further performance improvements become limited. After comparing the performance of the two type VLMs and taking inference efficiency into account, we ultimately select Qwen2.5VL-3B as the base VLM.

| VLM types&parameters | $x$-error | $y$-error | $\phi$-error |
|---|---|---|---|
| InternVL2.5-2B | 6.59m | 2.79m | 11.20° |
| InternVL2.5-4B | 3.71m | 1.44m | 4.89° |
| Qwen2.5VL-3B | 3.76m | **1.08m** | **3.80°** |
| Qwen2.5VL-7B | **3.60m** | **1.08m** | 3.83° |

Table 4: Mean absolute relative errors in key point predictions by the VLM under different model types and parameter sizes.

## 6 CONCLUSION

In this paper, we propose BIRDriver, a hierarchical framework that integrates a VLM with a motion planner. By leveraging the VLM's general-purpose understanding, high-level driving intentions are transmitted to the motion planner via key points. To enhance the accuracy of key point prediction, we introduce an auxiliary dataset along with a weighted SFT loss. Extensive experiments demonstrate

that BIRDriver achieves superior performance in long-tail scenarios, underscoring the effectiveness of integrating VLM with motion planner through key point guidance.

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

# A  INSTRUCTION FOR AUTONOMOUS DRIVING ASSISTANT

## A.1  SYSTEM PROMPT

You are an advanced autonomous driving assistant. Your role is to analyze Bird's Eye View (BEV) maps and carry out autonomous driving tasks.

**CORE PRINCIPLES**
**SAFETY FIRST**
• Always maintain a safe distance from other road users.
• Anticipate potential hazards and plan defensive paths.
• Adhere strictly to traffic laws and signals.
• Pay special attention to vulnerable road users (pedestrians, cyclists).

**PASSENGER COMFORT**
• Provide smooth lateral transitions and avoid sudden movements.
• Adjust lateral paths in a natural, human-like manner.
• Ensure proper lateral positioning across varying road conditions.
• Execute lane changes and turns gradually.

**EFFICIENCY**
• Optimize lateral movements without compromising safety.
• Limit unnecessary lane changes.
• Preserve adequate lateral clearance.
• Balance direct routes with smooth transitions.

**BEV MAP ELEMENTS:**
• Blue shaded areas: Drivable lanes (lighter blue indicates the planned route).
• Gray dashed lines: Lane center lines.
• Traffic signal lines (colored):
  – Green: Proceed
  – Yellow: Exercise caution
  – Red: Come to a stop
• Stoplines: Represent traffic light information on the planned route (red, yellow, or green areas).
• Objects:
  – Orange box with arrow: Ego vehicle (self).
  – Blue boxes with arrows: Other vehicles.
  – Brown boxes: Pedestrians.
  – Red boxes: Cyclists.
  – Gray hatched areas: Crosswalks.
  – Green solid lines: Historical 2-second motion trajectory of objects.
• Purple arrows: Reference path indicators.
• Gold star: Mission goal/destination.

**SPEED & ACCELERATION CONSTRAINTS:**
• Lateral acceleration $\leq$ 2 m/s².
• Minimize lateral jerk for comfort.
• Typical lane change duration: 3–5 seconds.

**SAFETY MARGINS:**
• Minimum lateral clearance: 1.5 meters.
• Increase margins in adverse conditions.
• Provide extra buffer for vulnerable road users.

**ERROR HANDLING:**
• If critical information is missing or unclear, always prioritize safety.
• When in doubt, assume the most conservative interpretation of other objects' intentions.
• Maintain a fail-safe lateral position in all scenarios.

## A.2 USER PROMPT

### A.2.1 INSTRUCTIONS FOR GENERATING KEY POINTS

Variant 1

<image>Based on the provided BEV map image, analyze the driving scenario and determine the lateral intention:
Provide the trajectory keypoints in the following format:
**[[x1, y1, $\theta$1], [x2, y2, $\theta$2], ..., [xn, yn, $\theta$n]]**

**Where:**
- x: longitudinal position in meters (relative to ego vehicle)
- y: lateral position in meters (relative to ego vehicle)
- theta: Heading angle in degrees, positive for left turns, negative for right turns.
- n: total number of keypoints ($n \leq 3$)

Each keypoint should represent a significant lateral movement intention, considering:
- Lane changes
- Turn preparations
- Obstacle avoidance maneuvers
- Path alignment adjustments

**Notes:**
- If there's no solid green track behind the vehicle, indicating it's stationary.
- The ego vehicle is shown as the orange box with an arrow.
- Prioritize safety (collision avoidance) and autonomy (self-recovery from blocked paths) in your planning.

Variant 2

<image>Predict the ego vehicle's immediate lateral driving path based on the BEV map scenario.
Output the planned trajectory as keypoints using this format:
**[[x1, y1, $\theta$1], [x2, y2, $\theta$2], ..., [xn, yn, $\theta$n]]**

**Details:**
- Coordinates (x, y) are relative to the ego vehicle in meters (x: forward, y: left).
- Heading angle (theta) is in degrees (positive left, negative right).
- Maximum 3 keypoints ($n \leq 3$).
- Keypoints should mark critical lateral actions like lane changes, turns, or avoidance.
- Follow the reference path (purple arrows) as the default, deviating only when necessary for safety or maneuvers, then smoothly return.
- Ensure the path is safe and avoids collisions.

Variant 3

<image>Analyze the BEV map. Plan a safe lateral maneuver sequence for the ego vehicle (orange box).
Provide the sequence as trajectory keypoints:
**[[x1, y1, $\theta$1], ..., [xn, yn, $\theta$n]]**, max 3 points.

**Parameters:**
- x (meters): forward distance from ego.
- y (meters): lateral distance from ego (left positive).
- theta (degrees): heading relative to ego's current heading (left positive).

**Instructions:**
- Prioritize collision avoidance above all.
- Follow the purple reference path unless a deviation (lane change, obstacle avoidance) is required.
- Ensure smooth realignment with the reference path after any deviation.
- Keypoints should capture the essence of the lateral plan.

Variant 4

> <image>Evaluate the current driving situation from the BEV map. Decide the optimal lateral course of action for the ego vehicle.
> Represent this decision as a sequence of trajectory keypoints:
> **[[x1, y1, $\theta$1], [x2, y2, $\theta$2], ..., [xn, yn, $\theta$n]]** ($n \leq 3$)
>
> **Coordinate System & Constraints:**
> - x: longitudinal (m), y: lateral (m), relative to ego.
> - theta: heading change (deg), positive left.
> - Must generally follow the reference path (purple arrows).
> - Deviate only for essential maneuvers (lane change, avoidance).
> - Prioritize safety and smooth recovery to the intended path.
> - Keypoints define significant lateral shifts or adjustments.

### A.2.2 INSTRUCTIONS FOR GENERATING COORDINATE POINTS

Variant 1

> <image>Based on the provided BEV map image, identify the three vehicles labeled with numbers 1, 2, and 3.
>
> Determine and output their longitudinal position (x), lateral position (y), and heading angle ($\theta$) relative to the ego vehicle, strictly in the order of their labels (1, then 2, then 3).
>
> Provide the information in the following format:
> **[[x1, y1, $\theta$1], [x2, y2, $\theta$2], [x3, y3, $\theta$3 ]]**
>
> **Where:**
> - x: longitudinal position in meters (relative to ego vehicle)
> - y: lateral position in meters (relative to ego vehicle)
> - $\theta$: Heading angle in degrees, positive for left turns, negative for right turns.

Variant 2

> <image>From the given BEV scene representation, extract the state parameters for the vehicles labeled 1, 2, and 3.
>
> For each vehicle, provide its relative x (forward/backward), y (left/right), and $\theta$ (heading) values referenced from the ego vehicle. Ensure the output follows the numerical order of the labels (1, 2, 3).
>
> Format the output strictly as:
> **[[x1, y1, $\theta$1], [x2, y2, $\theta$2], [x3, y3, $\theta$3]]**
>
> **Parameter definitions:**
> - x: Relative longitudinal position (m).
> - y: Relative lateral position (m).
> - $\theta$: Relative heading angle (deg), positive left turn.

Variant 3

> <image>Analyze the provided Bird's-Eye View (BEV) perception image. Locate the three distinct targets marked with numerical labels 1, 2, and 3.
>
> For each target, compute its pose relative to the ego vehicle's coordinate system. Specifically, determine its longitudinal distance (x), lateral distance (y), and yaw angle ($\theta$).
>
> Output the results sequentially according to the target labels (1 first, then 2, then 3) using the exact format below:
> **[[x1, y1, $\theta$1], [x2, y2, $\theta$2], [x3, y3, $\theta$3]]**

> **Where:**
> - x: Longitudinal distance (meters) from the ego vehicle origin.
> - y: Lateral distance (meters) from the ego vehicle origin.
> - $\theta$: Yaw angle (degrees), where counter-clockwise rotation relative to the ego vehicle's forward direction is positive.

Variant 4

> <image>Consider the ego vehicle as the origin (0,0) with its heading aligned with the positive x-axis in this BEV map. Identify the traffic participants labeled 1, 2, and 3.
>
> Determine the ego-centric coordinates (x, y) and relative orientation ($\theta$) for each labeled participant.
>
> Structure the output as a nested list containing the [x, y, $\theta$] for participant 1, followed by participant 2, and finally participant 3:
> **[[x_participant1, y_participant1, $\theta$_participant1],**
> **[x_participant2, y_participant2, $\theta$_participant2],**
> **[x_participant3, y_participant3, $\theta$_participant3]]**
>
> **Specifications:**
> - x, y units: meters.
> - $\theta$ units: degrees (positive counter-clockwise/left).

### A.2.3 INSTRUCTIONS FOR SCENE TYPE RECOGNITION

> **Instruction Variants (The following prefixes are used with the shared list below):**
>
> **Variant 1:**
> <image>Based on the BEVMap shown in the image, please determine the current driving scenario of the autonomous vehicle. Choose the most appropriate scenario from the following list of 57 predefined categories:
>
> **Variant 2:**
> <image>From the information in the BEVMap image, infer what type of driving situation the self-driving car is currently in. Use one of the 57 predefined scenario categories listed below:
>
> **Variant 3:**
> <image>Given the BEVMap in the image, identify which of the 57 labeled driving scenarios best describes the current situation of the autonomous vehicle. Here are all possible categories:
>
> **Variant 4:**
> <image>Please analyze the BEVMap provided in the image and select the most appropriate driving scenario type (from a set of 57 categories) that describes the present condition of the autonomous car:
>
> ---
>
> **Shared Category List (Applies to all variants above):**
> ```
> accelerating_at_crosswalk
> accelerating_at_traffic_light_with_lead
> behind_bike
> behind_pedestrian_on_driveable
> behind_pedestrian_on_pickup_dropoff
> changing_lane_to_left
> changing_lane_to_right
> changing_lane_with_lead
> crossed_by_bike
> crossed_by_vehicle
> following_lane_with_lead
> following_lane_with_slow_lead
> following_lane_without_lead
> high_lateral_acceleration
> high_magnitude_jerk
> ```

```
high_magnitude_speed
low_magnitude_speed
medium_magnitude_speed
near_barrier_on_driveable
near_construction_zone_sign
near_high_speed_vehicle
near_long_vehicle
near_multiple_pedestrians
near_multiple_vehicles
near_pedestrian_on_crosswalk
near_pedestrian_on_crosswalk_with_ego
near_trafficcone_on_driveable
on_carpark
on_intersection
on_pickup_dropoff
on_stopline_crosswalk
on_stopline_stop_sign
on_stopline_traffic_light
on_traffic_light_intersection
starting_protected_cross_turn
starting_protected_noncross_turn
starting_right_turn
starting_straight_stop_sign_intersection_traversal
starting_straight_traffic_light_intersection_traversal
starting_unprotected_cross_turn
starting_unprotected_noncross_turn
stationary
stationary_at_crosswalk
stationary_at_traffic_light_with_lead
stationary_at_traffic_light_without_lead
stationary_in_traffic
stopping_at_crosswalk
stopping_at_stop_sign_no_crosswalk
stopping_at_stop_sign_without_lead
stopping_at_traffic_light_without_lead
stopping_with_lead
traversing_crosswalk
traversing_intersection
traversing_narrow_lane
traversing_pickup_dropoff
traversing_traffic_light_intersection
waiting_for_pedestrian_to_cross
```

# B  IMPLEMENTATION DETAILS

**Dataset and Simulator.**    The nuPlan platform is a large-scale, closed-loop system for autonomous driving trajectory planning, comprising over 1,500 hours of expert driving data collected from four cities. It features a wide range of complex scenarios, including car following, lane changes, turns, intersections, bus stops, roundabouts, and pedestrian interactions. The platform includes a simulator that initializes from real-world scenarios. During simulation, non-ego agents are controlled either by log replay (non-reactive) or an Intelligent Driver Model policy (reactive), while the ego vehicle executes user-defined planned trajectories. Each 15-second simulation runs at 10 Hz, with a Linear Quadratic Regulator controller used to track the trajectory and issue control commands.

**Training Details.**    We adopted the LoRA technique for parameter-efficient fine-tuning of the VLM. Specifically, we set the LoRA rank to 16, the scaling factor to 32, and the dropout rate to 0.05, applying LoRA adapters to all linear layers. The part of language model was unfrozen during training. For optimization, we used the AdamW optimizer with an initial learning rate of $1 \times 10^{-4}$ and a weight decay of 0.1. The learning rate was scheduled using a cosine annealing strategy. Training was conducted on eight NVIDIA H800 GPUs over five epochs, taking approximately three days to complete. The proportions of the three dataset categories, namely *Key Point*, *Spatial Localization*,

and *Driving Scene Stepwise*, are 10:1:2. The entire dataset contains 838,824 samples. During the process of fine-tuning the motion planner, we followed the PLUTO framework to construct 1 million data splits. Training was conducted on eight RTX 4090 GPUs with a batch size of 128 over 10 epochs, taking approximately 6 hours to complete. We employed the AdamW optimizer with an initial learning rate of $1 \times 10^{-4}$ and a weight decay of $1 \times 10^{-4}$. Unlike PLUTO, however, we did not employ the warm-up technique.

## C    LIMITATIONS

Despite the numerous advantages of the hierarchical framework that combines a VLM with a motion planner, the inference efficiency of the VLM remains a key concern. In offline or simulation-based settings, where our method can be used to generate data or serve as a smart agent, this limitation is generally acceptable. For onboard deployment, the inference burden can be alleviated through techniques such as model quantization, asynchronous inference, or training a lightweight model to identify long-tail scenarios. We leave the exploration of real-world deployment strategies to future work.

## D    LLM STATEMENT

Large Language Models (LLMs) were employed solely for language refinement in this paper. Specifically, we used them to polish grammar, improve clarity, and enhance the academic style of our writing. The role of LLMs was limited to editing and improving the presentation of the text, without contributing to the technical content.

