# OpenReview forum: "Bird's-eye-view Informed Reasoning Driver"
_ICLR.cc/2026/Conference — ICLR 2026 Poster_

### Official Review · Reviewer_ctTg · 2025-10-29

**Soundness:** 3
**Presentation:** 3
**Contribution:** 3
**Rating:** 6
**Confidence:** 5

**Summary:**

This work proposes improvement over PLUTO hybrid planner by utilizing extra features generated by a VLM using BEV image. They fine tune the VLM to produce key points from BEV scene image using LoRA. These key points summarizes the trajectories of the agents in a compressed way. These are fed into the point encoding module of PLUTO planner to generate the trajectory. Their experiments show improvement on NuPlan and InterPlan over PLUTO.

**Strengths:**

1. BEV representation is quite generic and it's generalizable over sensor types, orientation and platforms. This can be applicable to a lot of different architectures and vehicle platforms.
2. Using key points, instead of dense trajectory representation, is more efficient.
3. Using weighted SFT loss and spatial localization learning for BEV are also generalizable techniques, that can benefit many works.

**Weaknesses:**

1. The approach is generalizable, but it's only shown with PLUTO planner. It would have made the case stronger if the authors also showed benefit with at least one more planner with BEV based feature augmentation.
2. The results are shown only using nuPlan, not using reactive simulator like CARLA. This work would benefit from adopting Bench2Drive for benchmarking.

**Questions:**

1. Key point generation could be possible from deterministic or classical learning methods as well, instead of using VLM. That would make the inference latency more practical. Did the authors consider this?
2. As the authors mention the PDCE loss, did they perform any comparison with their weighted SFT loss?

---

> ### Author Response · Authors · 2025-11-22
> **Re: the Reviewer ctTg**
>
> We thank you for the careful reading of our paper and constructive comments in detail.
>
> # 1. Change the base planner （Weakness1）
> We sincerely thank you for this constructive suggestion. We have supplemented our evaluation with additional experiments using PlanTF as the baseline motion planner. As shown in Table 1, BIRDriver (PlanTF [1]) consistently improves performance across the Test14-random, Test14-hard, and InterPlan benchmarks. **This demonstrates the strong transferability of our proposed method.**
>
> Table 1 Comparison of various planners across the Test14-random, Test14-hard, and InterPlan datasets.
> | Method            | Test14-random CLS-NR | Test14-random CLS-R | Test14-hard CLS-NR | Test14-hard CLS-R | InterPlan CLS-R |
> | ----------------- | -------------------- | ------------------- | ------------------ | ----------------- | --------------- |
> | PlanTF            | 85.98                | 80.53               | 70.10              | 61.49             | 36.53           |
> | BIRDriver(PlanTF) | 87.12                | 81.93               | 73.23              | 69.12             | 50.62           |
>
> # 2. Using reactive simulator (Weakness2)
> Thank you for this valuable suggestion. We fully agree that reactive simulation is crucial for accurately evaluating the driving capabilities of autonomous systems. We would like to clarify that the **nuPlan CLS-R (Closed-Loop Reactive) metric used in our experiments functions as a reactive simulator rather than a static log replay.** Specifically, surrounding agents adapt their behaviors in response to the ego vehicle's actions, enabling dynamic multi-agent interactions. In this regard, nuPlan CLS-R shares the same fundamental objective as CARLA [2]: assessing interactive driving performance to overcome the limitations of open-loop evaluation. While we acknowledge the merits of Bench2Drive, we focused on nuPlan in this study due to time and resource constraints.
>
> # 3. Methods of generating key points (Question1)
> We appreciate your meaningful suggestion regarding inference efficiency. Indeed, classical learning methods offer lower latency. However, **the core motivation of BIRDriver is to leverage the rich "common sense" and open-world knowledge embedded in VLMs**, rather than merely performing geometric regression for key points. Classical methods, even with large-scale training, are constrained by the specific data distribution and often **suffer from poor generalization in long-tail or unseen scenarios.** In contrast, our VLM-based approach enables the system to reason about complex, out-of-distribution contexts, ensuring higher robustness and safety. Therefore, we believe the latency trade-off is justified by the superior generalization capability.
>
>
> # 4. Compared with other loss functions (Question2)
> Thank you for the insightful asking. We have conducted additional experiments to compare the PDCE loss with our Weighted SFT loss. As shown in Table 2, our Weighted SFT loss achieves better prediction accuracy compared to PDCE.
>
> Table 2 Mean absolute relative errors in key point predictions by Qwen2.5VL-3B after fine-tuning on different loss function.
> | loss function                         | x-error | y-error | φ-error |
> | ------------------------------------- | ------- | ------- | ------- |
> | baseline with normal SFT loss         | 4.22m   | 1.19m   | 4.13°   |
> | baseline with PDCE loss               | 4.01m   | 1.15m   | 3.91°   |
> | baseline with Weighted SFT loss (Our) | **3.27m**   | **1.08m**   | **3.80°**   |
>
>
>
>
> # Reference
> [1] Cheng, Jie, et al. "Rethinking imitation-based planners for autonomous driving." 2024 IEEE International Conference on Robotics and Automation (ICRA). IEEE, 2024.
>
> [2] Dosovitskiy, Alexey, et al. "CARLA: An open urban driving simulator." Conference on robot learning. PMLR, 2017.

---

> ### Author Response · Authors · 2025-11-26
>
> Thank you again for the great efforts and valuable comments. We have carefully addressed the main concerns in detail. We hope you might find the response satisfactory. As the discussion phase is about to close, we are very much looking forward to hearing from you about any further feedback. We will be very happy to clarify any further concerns (if any).

---

### Official Review · Reviewer_ScDp · 2025-11-01

**Soundness:** 2
**Presentation:** 3
**Contribution:** 2
**Rating:** 4
**Confidence:** 4

**Summary:**

This paper proposes a hierarchical motion planning framework that integrates a Vision-Language Model (VLM) with a motion planner. Specifically, the VLM interprets the driving scene from a BEV map and outputs keypoints to represent high-level driving intentions. The motion planner PLUTO is then extended to incorporate these keypoints for trajectory planning.

**Strengths:**

1. The VLM in BIRDriver requires only the BEV map as input to convey the complete driving scene, eliminating the need for textual descriptions.

2. BIRDriver expresses driving intentions through keypoints, which offer advantages over meta-action, hidden-state, and waypoint-based approaches.

3. BIRDriver introduces a weighted SFT loss and three specialized datasets to enhance the accuracy of keypoint prediction.

4. BIRDriver achieves state-of-the-art performance in closed-loop evaluations on long-tail scenarios of the InterPlan benchmark.

**Weaknesses:**

1. The contribution of this work is limited: using BEV maps as VLM inputs has already been explored in prior work [1], and some studies [4] have even gone further by feeding BEV features directly into LLMs for autonomous driving tasks. Moreover, the authors claim that relying solely on BEV maps eliminates the need to align heterogeneous sensor data across different vehicle platforms. In fact, many existing works [2,3,4,5] have already achieved the same goal by using only map features or multi-view images as inputs.

2. The paper conducts experiments only on nuPlan and its long-tail test set InterPlan, lacking evaluation on broader datasets such as NuScenes and Waymo. If time and computational resources permit, extending experiments to more datasets and comparing with additional VLM-based autonomous driving methods[2,4,5,6,7,8,9,10] would be valuable.

[1] PlanAgent: A Multi-modal Large Language Agent for Closed-loop Vehicle Motion Planning
[2] Senna: Bridging Large Vision-Language Models and End-to-End Autonomous Driving
[3] Asynchronous Large Language Model Enhanced Planner for Autonomous Driving
[4] Holistic Autonomous Driving Understanding by Bird’s-Eye-View Injected Multi-Modal Large Models
[5] OmniDrive: A Holistic Vision-Language Dataset for Autonomous Driving with Counterfactual Reasoning
[6] VLP: Vision Language Planning for Autonomous Driving
[7] Making Large Language Models Better Planners with Reasoning-Decision Alignment
[8] VLM-AD: End-to-End Autonomous Driving through Vision-Language Model Supervision
[9] EMMA: End-to-End Multimodal Model for Autonomous Driving

[10] DRIVEVLM: The Convergence of Autonomous Driving and Large Vision-Language Models

**Questions:**

1. The paper extends the PLUTO architecture to implement the hierarchical framework. Can the proposed architecture be adapted to other existing advanced motion planners as well?

2. Can the proposed framework meet the real-time requirements of autonomous driving motion planning, given that VLM inference may be relatively slow?

---

> ### Author Response · Authors · 2025-11-22
> **(1/2) Re: the Reviewer ScDp**
>
> We thank you for the careful reading of our paper and constructive comments in detail.
>
> # 1. Concerns about the use of BEV maps as inputs is whether novel (Weakness1)
> We agree that using BEV maps as inputs to VLM/LLM-based driving systems has appeared in prior work. However, our contribution is not merely “using BEV,” but introducing a BEV-standardized, hierarchical closed-loop planner. Specifically, we have the following contributions:
>
> 1. **BEV-only hierarchical planner.** Unlike PlanAgent [1], which feeds an MLLM with both a BEV map and lane-graph textual descriptions, our planner carries all scene semantics solely by a single-frame BEV map without any scene-specific text. PlanAgent’s environment transformation explicitly constructs text inputs alongside BEV. Moreover, BEV-InMLLM [2] targets multi-view/temporal driving QA and injects BEV features derived from multi-view videos for holistic understanding, rather than serving as a single-frame BEV closed-loop planning interface.
>
> 2. **Sparse intention interface (≤3 relative (x, y, ϕ) keypoints).** Our VLM outputs at most three relative geometric keypoints, adaptively extracted via RDP, as a minimal sufficient high-level intention. **This is fundamentally different from natural-language meta-actions in Senna [3], hidden-state/instruction guidance in AsyncDriver [4], or dense waypoint generation in DriveVLM [5].** This interface explicitly decouples what VLMs are good at (strategic geometric intent) from what real-time planners are good at (high-frequency trajectory refinement).
>
> 3. **Precision-oriented tuning for reliable numeric intent generation.** To make sparse keypoint tokens accurate and stable, we introduce **three auxiliary datasets and a token-weighted SFT strategy**, enabling precise numerical coordinate generation. This leads to state-of-the-art closed-loop long-tail performance on InterPlan, and consistent improvements over the base planner on Test14-hard, indicating a genuine robustness gain rather than an input repackaging effect.
>
> Regarding platform-agnostic deployment, prior works can also improve transferability by using map features or multiview images. Our point is more practical: **by consuming the production-ready BEV stream already output by modern perception stacks, our planning module relies on a standardized intermediate representation rather than raw sensor layouts**, making cross-platform deployment more straightforward and reducing per-vehicle alignment effort.
>
> # 2. Concerns about the choice of datasets and baselines (Weakness2)
> We sincerely thank you for this valuable suggestion. We acknowledge that evaluating on a broader range of datasets would further validate the generalizability of our method. However, regarding our choice of datasets and baselines, we would like to clarify the rationale behind our experimental design:
>
> 1. **The Necessity of Closed-Loop Evaluation:** Our research focuses on reactive planning and interaction, which necessitates a high-fidelity closed-loop simulation environment. nuPlan [6] is currently the standard benchmark for this purpose. It provides reactive agents and specific planning metrics (e.g., drivability, comfort, progress) that rigorously assess how a planner handles dynamic interactions over long horizons.
>
> 2. **Limitations of Open-Loop Evaluation:** Traditional benchmarks like Waymo [7] and nuScenes [8] primarily focus on open-loop evaluation. As noted in recent literature [9-10], open-loop metrics (e.g., L2 error or collision rate) **often fail to capture true driving quality because the ego vehicle's actions do not influence the future states of surrounding agents.**
>
> 3. **Bridging the "Sim-to-Real" Gap:** It is well established that models performing well in open-loop (non-reactive) settings do not necessarily translate to better driving in closed-loop (reactive) environments. Therefore, we prioritized nuPlan to rigorously test the **interactive planning capability** of our VLM-based method, rather than merely its trajectory fitting ability.
>
> 4. **Comparison with Baselines:** Unlike several of the open-loop works cited, BIRDriver targets mid-to-mid / BEV-only deployable hierarchical planning specifically within the nuPlan closed-loop benchmark. We have conducted extensive comparisons with **PlanAgent[1] and InstructDriver[12], which share this closed-loop setting, and have demonstrated superior performance.**
>
> Finally, while we have not conducted tests on the Waymo dataset due to time and resource constraints, we understand your concern. If you believe that open-loop evaluation on Waymo is essential for validating our contributions, we will do our best to include these additional experiments.

---

> ### Author Response · Authors · 2025-11-22
> **(2/2) Re: the Reviewer ScDp**
>
> # 3. Change the base planner (Question1)
> We sincerely thank you for this constructive suggestion. We have supplemented our evaluation with additional experiments using PlanTF as the baseline motion planner. As shown in Table 1, BIRDriver (PlanTF [13]) consistently improves performance across the Test14-random, Test14-hard, and InterPlan benchmarks. **This demonstrates the strong transferability of our proposed method.**
>
> Table 1 Comparison of various planners across the Test14-random, Test14-hard, and InterPlan datasets.
> | Method            | Test14-random CLS-NR | Test14-random CLS-R | Test14-hard CLS-NR | Test14-hard CLS-R | InterPlan CLS-R |
> | ----------------- | -------------------- | ------------------- | ------------------ | ----------------- | --------------- |
> | PlanTF            | 85.98                | 80.53               | 70.10              | 61.49             | 36.53           |
> | BIRDriver(PlanTF) | 87.12                | 81.93               | 73.23              | 69.12             | 50.62           |
>
> # 4. Concerns about inference efficiency and onboard deployment (Question2)
> We appreciate the your valid concern regarding real-time performance. We have successfully deployed Qwen2.5-VL-3B on a hardware platform with computing power similar to the NVIDIA Thor X platform [14]. Through the use of **multi-token prediction (MTP) [15] and low-bit quantization techniques, our real-car deployment achieves a VLM inference rate of ~5Hz**, paired with a **10Hz frequency for the ML planner**.
>
> To ensure safety and fluidity, we employ an **asynchronous inference pipeline**, a method validated by works such as AsyncDriver [4], to achieve real-time vehicle control. Additionally, unlike methods that generate complete trajectories [5], **BIRDriver is optimized to output sparse coordinates (up to 3 keypoints)**, which are less prone to latency jittering compared to trajectories.
>
>
> # Reference
> [1] Zheng, Yupeng, et al. "Planagent: A multi-modal large language agent for closed-loop vehicle motion planning." arXiv preprint arXiv:2406.01587 (2024).
>
> [2] Ding, Xinpeng, et al. "Holistic autonomous driving understanding by bird's-eye-view injected multi-modal large models." Proceedings of the IEEE/CVF Conference on Computer Vision and Pattern Recognition. 2024.
>
> [3] Jiang, Bo, et al. "Senna: Bridging large vision-language models and end-to-end autonomous driving." arXiv preprint arXiv:2410.22313 (2024).
>
> [4] Chen, Yuan, et al. "Asynchronous large language model enhanced planner for autonomous driving." European Conference on Computer Vision. Cham: Springer Nature Switzerland, 2024.
>
> [5] Tian, Xiaoyu, et al. "Drivevlm: The convergence of autonomous driving and large vision-language models." arXiv preprint arXiv:2402.12289 (2024).
>
> [6] Caesar, Holger, et al. "nuplan: A closed-loop ml-based planning benchmark for autonomous vehicles." arXiv preprint arXiv:2106.11810 (2021).
>
> [7] Sun, Pei, et al. "Scalability in perception for autonomous driving: Waymo open dataset." Proceedings of the IEEE/CVF conference on computer vision and pattern recognition. 2020.
>
> [8] Caesar, Holger, et al. "nuscenes: A multimodal dataset for autonomous driving." Proceedings of the IEEE/CVF conference on computer vision and pattern recognition. 2020.
>
> [9] Fan, Yixuan, Yali Li, and Shengjin Wang. "Risk-Aware Self-consistent Imitation Learning for Trajectory Planning in Autonomous Driving." European Conference on Computer Vision. Cham: Springer Nature Switzerland, 2024.
>
> [10] Bouzidi, Mohamed-Khalil, et al. "Closing the loop: Motion prediction models beyond open-loop benchmarks." arXiv preprint arXiv:2505.05638 (2025).
>
> [12] Zhang, Ruijun, et al. "Instruct large language models to drive like humans." arXiv preprint arXiv:2406.07296 (2024).
>
> [13] Cheng, Jie, et al. "Rethinking imitation-based planners for autonomous driving." 2024 IEEE International Conference on Robotics and Automation (ICRA). IEEE, 2024.
>
> [14] https://developer.nvidia.com/drive/agx#section-thor-features
>
> [15] Gloeckle, Fabian, et al. "Better & faster large language models via multi-token prediction." Proceedings of the 41st International Conference on Machine Learning. 2024.

---

> ### Author Response · Authors · 2025-11-26
>
> Thank you again for the great efforts and valuable comments. We have carefully addressed the main concerns in detail. We hope you might find the response satisfactory. As the discussion phase is about to close, we are very much looking forward to hearing from you about any further feedback. We will be very happy to clarify any further concerns (if any).

---

### Official Review · Reviewer_BksK · 2025-11-01

**Soundness:** 2
**Presentation:** 2
**Contribution:** 2
**Rating:** 4
**Confidence:** 4

**Summary:**

This paper proposes BIRDriver, an hierarchical framework that integrates Vision-Language Models (VLMs) with a motion planner to address the motion planning challenges in complex, long-tail autonomous driving scenarios. Unlike existing methods, BIRDriver utilizes only a single-frame Bird's-Eye View (BEV) as the primary input for the VLM, avoiding the reliance on domain-specific encoders and the complexity of multi-modal data alignment. The framework leverages the VLM's common-sense reasoning capabilities to generate keypoints, which are subsequently passed to the motion planner to produce the final trajectory. To address the difficulty of standard VLMs in generating precise coordinates, the authors designed three auxiliary datasets and a token-level weighting mechanism, significantly improving keypoint prediction accuracy.

**Strengths:**

1、VLM Integration Paradigm: The paper introduces a hierarchical architecture where a Vision-Language Model (VLM) generates high-level keypoints, while a conventional planner executes low-level trajectories. This approach elegantly balances the advanced reasoning capabilities of VLMs with the real-time performance and safety of traditional planners.

2、Dataset Innovation: To address the limitations of Vision-Language Models (VLMs) in spatial localization and scene understanding, the authors meticulously designed three auxiliary datasets. This multi-task learning strategy significantly improved the accuracy of keypoint prediction.

3、Comprehensive Experimental Validation: The method was extensively validated on the nuPlan dataset across three benchmarks: Test14-random, Test14-hard, and InterPlan. Notably, on the InterPlan long-tail test set, BIRDriver significantly outperforms PLUTO and Diffusion Planner, demonstrating its effectiveness in handling complex and rare scenarios.

**Weaknesses:**

1. BEV-Only Input: Relying solely on BEV as input is overly idealistic and impractical. While the dataset currently in use contains BEV information, other datasets may not. More critically, ground-truth BEV information is not available during real-world driving. This would necessitate the model to predict the BEV, which in turn would introduce errors. Such a practical setting is far more complex than one with standard multi-view images, raising concerns about the model's generalizability and practical deployability.

2. Inference Efficiency: The authors' acknowledgement of the VLM's inference efficiency issues in the limitations section is commendable for its honesty. A clarification is needed: What is the inference time for a single VLM call? Motion planning typically requires high-frequency operation (e.g., 10Hz or higher). Is the current framework executed synchronously (with the VLM being called at every step) or asynchronously? If synchronous, the VLM's latency would directly impact the system's reaction time.

3. Keypoints: While the authors advocate for using no more than three keypoints, there is a lack of sufficient theoretical and experimental justification for why three is the optimal number. The experiment in Section 5.3 only compares the RDP algorithm with a method using only the final trajectory point, failing to explore the performance differences among using 2, 3, or 4 keypoints. In certain highly complex scenarios, three keypoints may be insufficient to fully express the complete driving intent, yet the paper does not investigate the relationship between the number of keypoints and performance. Additionally, more details regarding the setting of ε are required.

4. Minor Performance Degradation in Simple Scenarios and Insufficient Analysis: As shown in Table 1, on the Test14-random benchmark, BIRDriver (91.46) performs slightly worse than its base model, PLUTO (91.87), on the CLS-NR metric. Although the margin is small, this raises an important question: Does the intervention of the VLM introduce unnecessary complexity or superfluous decisions in simple, routine scenarios, thereby causing this slight performance degradation? Does this imply the need for an arbitration mechanism between the VLM and the base planner, which would activate the VLM only when a difficult scenario is detected? The authors should provide an analysis and discussion of this phenomenon.

**Questions:**

Refer to Weaknesses.

---

> ### Author Response · Authors · 2025-11-22
> **(1/3) Re: the Reviewer BksK**
>
> We thank you for the careful reading of our paper and constructive comments in detail.
>
> # 1. Concerns about reliance on "Ground Truth" (GT) BEV inputs (Weakness1)
> Thank you for the insightful comment regarding the reliance on "Ground Truth" (GT) BEV inputs. We strictly agree that a robust planning framework must handle the imperfections inherent in real-world perception systems, rather than relying on idealized data. To address your concern, we would like to clarify two critical points regarding the nature of our data and the robustness of our method:
>
> 1. **The "Ground Truth" in nuPlan is Derived from Real-World Sensors.** It is important to highlight that BIRDriver is trained and evaluated on the nuPlan dataset, which consists of over 1,500 hours of real-world expert driving data, not synthetic data generated by a simulated engine (e.g., CARLA [1]).
> - The "GT" trajectories and object bounding boxes in nuPlan are generated **via offline perception pipelines processing raw sensor data** [2]. Consequently, the input BEV maps inherently contain sensor noise, minor calibration errors, and geometric fluctuations typical of physical sensors.
> - Since our VLM is fine-tuned on this dataset, it has already learned to reason with these real-world artifacts, rather than relying on abstract, perfect geometric ideals.
>
>
>
> 2. **Robustness Analysis with Distance-Dependent Random Masking**. To evaluate the robustness of our model against sensor occlusions and limited perception ranges, we conducted an additional experiment involving Distance-Dependent Random Masking. Specifically, we randomly masked 20% of traffic participants located more than 20 meters from the ego vehicle. This setup simulates real-world scenarios where distant objects are prone to detection failures. We performed evaluations on the Test14-hard and InterPlan datasets, with results presented in Table 1. The results indicate that while random masking causes a marginal performance decline for BIRDriver, **our method still maintains a significant improvement over the baseline PLUTO.**
>
> Table 1 Performance under distance-dependent random masking. Bold indicates the best performance.
> | Method | **Test14-hard** CLS-NR | **Test14-hard** CLS-R | InterPlan CLS-R
> | :--- | :---: | :---: |:---: |
> | PLUTO | 80.03 | 76.92 |48.92 |
> | BIRDriver (PLUTO) | **80.56** | **80.33** |**55.29** |
> | BIRDriver-mask (PLUTO) | 80.41 | 80.20 |54.11 |
>
> ---
>
> Structurally, our bi-level system aligns with established industry standards. Leading industry players, including the Baidu Apollo [3], and Harmony Intelligent Mobility Alliance [4], employ this decoupled paradigm, where upstream perception produces unified BEV representations or structured object lists to drive downstream prediction and planning. **Distinguished by superior interpretability, this architecture remains a cornerstone of production-grade autonomous driving solutions.** BIRDriver functions as a dedicated planning module that consumes mid-level structured perception inputs rather than raw sensor data. We advance this standard by integrating a VLM to enable commonsense reasoning and address long-tail corner cases based on BEV maps, solving challenges that have persistently hindered traditional rule-based or learning-based planners.
>
>
> # 2. Concerns about inference efficiency and onboard deployment (Weakness2)
> We appreciate your valid concern regarding real-time performance. We have successfully deployed Qwen2.5-VL-3B on a hardware platform with computing power similar to the NVIDIA Thor X platform [5]. Through the use of **multi-token prediction (MTP) [6] and low-bit quantization techniques, our real-car deployment achieves a VLM inference rate of ~5Hz**, paired with a **10Hz frequency for the ML planner**.
>
> To ensure safety and fluidity, we employ an **asynchronous inference pipeline** in the actual vehicle deployment, a method validated by works such as AsyncDriver [7], to achieve real-time vehicle control. Additionally, distinct from approaches that generate complete trajectories [8], **BRIDriver is optimized to output sparse coordinates (maximum 3 keypoints)**, which are less prone to latency jittering compared to trajectories.

---

> ### Author Response · Authors · 2025-11-22
> **(2/3) Re: the Reviewer BksK**
>
> # 3. Concerns about the maximum number of keypoints $N$ and the details of $\epsilon$ (Weakness3)
> We sincerely thank you for this insightful question regarding the rationale behind setting the maximum number of keypoints $N$ and the details of $\epsilon$. We understand the concern about whether $N=3$ is sufficient for complex scenarios, and we address this design choice from two analytical perspectives:
>
> 1. In local motion planning, most vehicle maneuvers (e.g., lane changes, turns) can be mathematically approximated using cubic spline interpolation. A cubic curve is typically defined by four control points. Since the ego vehicle's current position and heading are already known (serving as the first fixed point), **the model only needs to predict three additional points (intermediate points + endpoint)** to mathematically constrain and reconstruct a complex, smooth curve. **Inspired by this kinematic property, we selected 3 as the upper limit for the number of key points.**
>
> 2. We conducted additional experiments on the InterPlan dataset, which contains numerous complex, untrained scenarios such as obstacle avoidance and stalled vehicles, comparing key point limits of $N=2$ and $N=4$. As shown in Table 2, **setting the limit to 2 yields performance inferior to 3, as it is insufficient to capture high-level decision-making intent. Conversely, the performance with a limit of 4 is comparable to that with 3.** Consequently, we determined that 3 is the standard setting balancing expressiveness and efficiency.
>
> Table 2 Performance of the InternVL2.5-4B model trained with different key point extraction methods on the InterPlan dataset.
> | Key point extraction methods   | InterPlan |
> | ------------------------------ | --------- |
> | Final trajectory point         | 34.72     |
> | RDP with 2 points (Our method) | 49.68     |
> | RDP with 3 points (Our method) | **53.81**     |
> | RDP with 4 points (Our method) | 53.65     |
>
> 3. Regarding the parameter setting, **we set the $\epsilon$ of RDP algorithm to 0.02.** We have updated this configuration in the Implementation Details section of the revised manuscript to improve clarity.
>
> # 4. Concerns about the performance trade-off in simple scenarios (Weakness4)
> Thank you for this keen observation regarding the performance trade-off in simple scenarios. We have conducted a comprehensive analysis of this phenomenon and offer the following clarification:
>
> On the Test14-random benchmark (comprising 261 scenarios), while BIRDriver shows a marginal decline of 0.41 points in the CLS-NR (Non-Reactive) metric compared to the base PLUTO model, **it achieves a notable improvement of 1.23 points in the CLS-R (Reactive) setting.**
>
> We attribute this to the intrinsic nature of the metrics. In the CLS-NR setting, where other agents do not interact with the ego vehicle, the optimal policy often involves strictly overfitting the logged expert trajectory. Even if the VLM outputs a "safe but slightly different" intermediate key point, it may cause the trajectory to deviate marginally from the specific expert log, resulting in a minor score deduction in the open-loop metric. However, **the performance gain in the CLS-R setting—where agents are reactive and the simulator mirrors real-world dynamics—demonstrates that the VLM intervention does not introduce "unnecessary complexity."** Instead, **it instills a more robust interactive driving policy that handles dynamic agents more effectively than the base model**, a critical capability in simulation that is not fully captured by Log Replay (NR). Therefore, we consider the negligible drop in CLS-NR an acceptable trade-off for superior reactive capability.
>
> Furthermore, to mitigate the potential negative impact of VLM prediction errors in simple scenarios, **we implemented a noise injection strategy during the fine-tuning of the motion planner, as detailed in Method 4.4.** By adding Gaussian noise to the ground-truth key points during training, we ensure that PLUTO becomes robust to input variations and focuses on adhering to the high-level driving intentions provided by the VLM.
>
> ---
>
> Regarding the suggestion of introducing an arbitration mechanism to selectively activate the VLM, **we contend that training a discriminator to accurately quantify "scenario complexity" is a non-trivial task that introduces new dependencies on recognition accuracy.** Such a mechanism could inadvertently become a performance bottleneck; if the discriminator fails to trigger the VLM in subtle long-tail scenarios, the framework would forfeit the critical benefits of the VLM’s reasoning capabilities. While we acknowledge that a gating mechanism is a valid and necessary approach for optimizing computational efficiency in engineering deployment—as noted in our Limitations section—**our current academic focus lies in demonstrating the VLM's intrinsic capability to guide planning directly, without interference from an external gatekeeper.**

---

> ### Author Response · Authors · 2025-11-22
> **(3/3) Re: the Reviewer BksK**
>
> # Reference
> [1] Dosovitskiy, Alexey, et al. "CARLA: An open urban driving simulator." Conference on robot learning. PMLR, 2017.
>
> [2] Caesar, Holger, et al. "nuplan: A closed-loop ml-based planning benchmark for autonomous vehicles." arXiv preprint arXiv:2106.11810 (2021).
>
> [3] https://www.apollo.auto
>
> [4] https://hima.auto
>
> [5] https://developer.nvidia.com/drive/agx#section-thor-features
>
> [6] Gloeckle, Fabian, et al. "Better & faster large language models via multi-token prediction." Proceedings of the 41st International Conference on Machine Learning. 2024.
>
> [7] Chen, Yuan, et al. "Asynchronous large language model enhanced planner for autonomous driving." European Conference on Computer Vision. Cham: Springer Nature Switzerland, 2024.
>
> [8] Tian, Xiaoyu, et al. "Drivevlm: The convergence of autonomous driving and large vision-language models." arXiv preprint arXiv:2402.12289 (2024).

---

> ### Author Response · Authors · 2025-11-26
>
> Thank you again for the great efforts and valuable comments. We have carefully addressed the main concerns in detail. We hope you might find the response satisfactory. As the discussion phase is about to close, we are very much looking forward to hearing from you about any further feedback. We will be very happy to clarify any further concerns (if any).

---

### Official Review · Reviewer_2yL1 · 2025-11-01

**Soundness:** 2
**Presentation:** 2
**Contribution:** 2
**Rating:** 6
**Confidence:** 4

**Summary:**

The paper proposes BIRDriver (Bird’s-eye-view Informed Reasoning Driver) that contains  hierarchical framework that integrates a Vision-Language Model (VLM) with a motion planner to improve autonomous driving to address long-tail scenarios. Unlike prior approaches that rely on multi-frame camera inputs or large waypoint sequences, BIRDriver uses a single-frame Bird’s-Eye-View (BEV) map and generates a small set of key points (≤3) to convey high-level driving intentions to the motion planner.
BIRDriver key contributions are:

- In BIRDriver, perception input tokens along with BEV maps are passed as input to VLM which interprets and outputs key points; motion planner generates trajectories based on these points.
- In BIRDrivier, it predicts relative key points small set of points instead of full trajectories for efficient intention transmission.
- BIRDriver uses three auxiliary datasets (Key Point, Spatial Localization, Driving Scene Stepwise) and a weighted SFT loss to improve numeric precision.

**Strengths:**

- BIRDriver combines VLM reasoning with BEV representation for motion planning, that results in reducing reliance on multi-modal alignment.
- In BIRDrivier, it predicts relative a small set of key points that are  compact, interpretable interface between reasoning and planning.
- In BIRDrivier, is good at spatial localization and scene understanding that addresses numeric token importance, improving precision in key point generation.

**Weaknesses:**

- BIRDriver latency or inference times remains still, it concerns whether BIRDriver can be applied to real-time applications: VLM inference of BIRDriver remains heavy for real-time onboard deployment; paper only briefly mentions quantization and asynchronous inference as future work.
- BIRDriver experiments are conducted on limited real-world validation: Experiments are simulation-based  Test14-random, Test14-hard, and InterPlan datasets. It would better to see zero-shot comparisons from simulation to real-world like nuScenes or waymo.
- Paper lacks detailed discussion on failure cases or robustness under sensor noise and occlusions.
- BIRDriver mostly focuses on nuPlan benchmarks; does not compare against recent RL-based or hybrid planning approaches like Diffusion Planner.
- BIRDriver highly relies on BEV maps omiting temporal context beyond 2-second history; may limit reasoning in highly dynamic scenes.

**Questions:**

- How does BIRDriver handle ambiguity in BEV maps, such as occluded agents or missing traffic signals?
- Could temporal context (multi-frame BEV) improve reasoning without significantly increasing complexity?
- How sensitive is the system to BEV map quality (e.g., sensor noise, map inaccuracies)?
- It would be better to see how weighted SFT loss improves compared to reinforcement learning  GRPO like in Diffusion Planner or geometry-aware objectives?
- What are the most common failure modes observed in InterPlan scenarios?
- How does the system perform under real-time constraints? Any latency benchmarks?

---

> ### Author Response · Authors · 2025-11-22
> **(1/3) Re: the Reviewer 2yL1**
>
> Thank you very much for the careful reading and constructive comments.
>
> # 1. Concerns about inference efficiency and onboard deployment (Weakness1 & Q6)
> We appreciate the your valid concern regarding real-time performance. We have successfully deployed Qwen2.5-VL-3B on a hardware platform with computing power similar to the NVIDIA Thor X platform [1]. Through the use of **multi-token prediction (MTP) [2] and low-bit quantization techniques, our real-car deployment achieves a VLM inference rate of ~5Hz**, paired with a **10Hz frequency for the ML planner**.
>
> To ensure safety and fluidity, we employ an **asynchronous inference pipeline**, a method validated by works such as AsyncDriver [3], to achieve real-time vehicle control. Additionally, unlike methods that generate complete trajectories [4], **BIRDriver is optimized to output sparse coordinates (up to 3 keypoints)**, which are less prone to latency jittering compared to trajectories.
>
>
> # 2. Concerns about real-world validation (Weakness2)
> Thank you for the insightful comment and we fully agree that real-world validation is essential. However, we would like to offer a clarification regarding the nature of the nuPlan benchmark used in our work, which might differ from traditional definitions of "simulation" (e.g., CARLA [5]).
>
> 1. nuPlan is built from large-scale real-world logs. Unlike synthetic simulators, the nuPlan dataset **contains 1,500+ hours of human driving data** collected across four cities (Boston, Pittsburgh, Las Vegas, and Singapore). Therefore, our model is **trained and evaluated on real-world sensor data and trajectories**, rather than synthetic data.
>
> 2. nuPlan uses closed-loop reactive evaluation. In the context of nuPlan, the term "simulation" refers to **closed-loop reactive evaluation**. This setting is **more challenging and closer to real-world deployment than the open-loop evaluation typically used with standard nuScenes [6] or Waymo [7] benchmarks**. In Open-loop (standard nuScenes ), the ego-vehicle's actions do not affect the environment. In Closed-loop (nuPlan), the system must plan trajectories over long horizons where deviations accumulate, and surrounding agents can be reactive. It's well known that models with better performance in open-loop (nonreactive) evaluations do not necessarily drive better in closed-loop (reactive) evaluations.
>
> Given that nuPlan is fundamentally constructed from real-world driving logs, BIRDriver’s strong performance in these rigorous, reactive scenarios serves as robust validation of its real-world applicability.
>
> # 4. Comparison with more algorithms (Weakness4)
> Thank you for this insightful comment and for highlighting the importance of comparing against diverse planning paradigms.
>
> We would like to respectfully clarify that **we have indeed included Diffusion Planner [8] as a primary baseline in our evaluation. Please refer to Table 1 (Row 7).** Our method demonstrates a significant advantage over Diffusion Planner, in the long-tail InterPlan benchmark (achieving a score of 55.29 vs. 39.85).
>
> Regarding other RL-based approaches, we note that the core contribution of BIRDriver is leveraging the semantic reasoning and commonsense knowledge of VLMs to solve long-tail cases (e.g., interpreting construction signs). While RL agents typically rely on extensive trial-and-error for trajectory optimization, our method aligns more closely with approaches that attempt to reason about the scene or clone human behavior. Therefore, we believe the most direct comparison is against agents like PLUTO, Diffusion Planner, PlanAgent and InstructDriver. Moreover, **the proposed method doesn't conflict with RL training, it's plausible and desirable to combine both aspects to achieve a great driving model.**
>
> # 5. Concerns about using only historical 2s data (Weakness5)
> We appreciate your concern regarding the utilization of a 2-second historical horizon. It is worth noting, however, that our experimental configuration **is consistent with prevailing SOTA methodologies, including PlanTF, PLUTO, and Diffusion Planner, all of which utilize a similar timeframe.**
>
> In highly dynamic environments, **decision-making is predominantly reactive and intuitive [9-10], placing greater emphasis on immediate context rather than extended historical sequences.** That said, we concur that leveraging Retrieval-Augmented Generation (RAG) to retrieve significant long-term historical patterns could offer valuable context for decision-making. We greatly appreciate this constructive suggestion, which points toward a valuable avenue for future research.

---

> ### Author Response · Authors · 2025-11-22
> **(2/3) Re: the Reviewer 2yL1**
>
> # 6. How to handle occluded agents or missing traffic signals（Question1 & Weakness3）
> Thank you for this question. Consistent with state-of-the-art methods such as PLUTO and Diffusion Planner, **BIRDriver does not employ additional explicit mechanisms to handle occluded agents or missing traffic signals.** This follows the standard paradigm of data-driven approaches, acknowledging that such ambiguities are often inevitable in real-world scenarios, and our model relies on learning from the data distribution to handle these uncertainties implicitly.
>
> # 7. Analysis of using multi-frame BEV as inputs (Question2)
> We appreciate your question regarding whether multi-frame BEV could enhance reasoning without adding substantial complexity. In practice, directly stacking multi-frame BEV maps **significantly increases the token count compared to single-frame inputs, making it difficult for the VLM to selectively attend to truly relevant temporal cues.** Processing such dense inputs typically requires additional architectural components [11] or temporal-specific objectives [12] **to prevent the model from focusing on redundant tokens rather than meaningful motion cues.** Furthermore, multi-frame BEV fusion introduces non-negligible computation and memory overhead; recent studies [13] explicitly note that naïve fusion significantly increases inference latency. Therefore, while temporal context can be beneficial, its integration is not without cost, and our design choices intentionally aim to avoid these prohibitive overheads while maintaining strong reasoning performance.
>
> # 8. The impact of BEVmap quality (Question3 & Weakness3)
> Thank you for this insightful comment regarding the robustness of our system in the presence of input imperfections. We address this concern from two perspectives: **the inherent nature of our training data and a dedicated stress-test experiment.**
>
> It is important to highlight that BIRDriver is trained and evaluated on the nuPlan dataset. The structured information we use to construct the **BEV Map is generated via offline perception pipelines processing raw sensor data, rather than relying on perfect, human-annotated ground truth.** Consequently, our training data inherently encapsulates sensor noise, minor calibration errors, and geometric fluctuations typical of real-world perception systems. This implies that BIRDriver has implicitly learned to be resilient to such imperfections during the standard training process.
>
> ---
>
> We further evaluated robustness against sensor noise by conducting a Distance-Dependent Random Masking experiment on the Test14-hard and InterPlan datasets, where 20% of agents beyond 20 meters were masked to simulate detection instability. As shown in the Table 1, while this setup leads to a marginal performance drop, BIRDriver exhibits remarkable stability. Crucially, even with this input degradation, our method (BIRDriver-mask) continues to significantly outperform the baseline PLUTO, confirming its resilience to perception inaccuracies.
>
> Table 1 Performance under distance-dependent random masking. Bold indicates the best performance.
> | Method | **Test14-hard**  CLS-NR | **Test14-hard**  CLS-R | InterPlan CLS-R
> | :--- | :---: | :---: |:---: |
> | PLUTO | 80.03 | 76.92 |48.92 |
> | BIRDriver (PLUTO) | **80.56** | **80.33** |**55.29** |
> | BIRDriver-mask (PLUTO) | 80.41 | 80.20 |54.11 |
>
> # 9. Compared with other loss functions (Question4)
> We sincerely thank you for this insightful suggestion.
>
> First, we would like to clarify that, **to the best of our knowledge, the Diffusion Planner does not explicitly employ GRPO for enhancing planning performance.** However, we agree that integrating RL-based optimization is a highly promising direction. Your suggestion has inspired us to include RL in our future work to further push the performance boundaries of our model.
>
> Furthermore, we conducted additional experiments comparing our Weighted SFT loss against the PDCE loss [14]. We evaluated the performance of the Qwen2.5VL-3B model after fine-tuning with these different loss functions. As shown in Table 2, our method demonstrates superior performance:
>
> Table 2 Mean absolute relative errors in key point predictions by Qwen2.5VL-3B after fine-tuning on different loss function.
> | loss function                  | x-error | y-error | φ-error |
> | ------ | ------- | ------- | ----- |
> | baseline with normal SFT loss         | 4.22m   | 1.19m   | 4.13°   |
> | baseline with PDCE loss       | 4.01m   | 1.15m   | 3.91°   |
> | baseline with Weighted SFT loss (Our) | **3.27m**   | **1.08m**   | **3.80°**   |
>
> # 10. Common failure modes in InterPlan (Question5 & Weakness3)
> While BIRDriver demonstrates robust performance in structured traffic scenarios, we observed that failure cases are primarily concentrated in unstructured interactions, specifically with jaywalking pedestrians. The behaviors of these agents are inherently difficult to anticipate due to their volatile movement patterns.

---

> ### Author Response · Authors · 2025-11-22
> **(3/3) Re: the Reviewer 2yL1**
>
> # References
> [1] https://developer.nvidia.com/drive/agx#section-thor-features
>
> [2] Gloeckle, Fabian, et al. "Better & faster large language models via multi-token prediction." Proceedings of the 41st International Conference on Machine Learning. 2024.
>
> [3] Chen, Yuan, et al. "Asynchronous large language model enhanced planner for autonomous driving." European Conference on Computer Vision. Cham: Springer Nature Switzerland, 2024.
>
> [4] Tian, Xiaoyu, et al. "Drivevlm: The convergence of autonomous driving and large vision-language models." arXiv preprint arXiv:2402.12289 (2024).
>
> [5] Dosovitskiy, Alexey, et al. "CARLA: An open urban driving simulator." Conference on robot learning. PMLR, 2017.
>
> [6] Caesar, Holger, et al. "nuscenes: A multimodal dataset for autonomous driving." Proceedings of the IEEE/CVF conference on computer vision and pattern recognition. 2020.
>
> [7] Sun, Pei, et al. "Scalability in perception for autonomous driving: Waymo open dataset." Proceedings of the IEEE/CVF conference on computer vision and pattern recognition. 2020.
>
> [8] Zheng, Yinan, et al. "Diffusion-based planning for autonomous driving with flexible guidance." arXiv preprint arXiv:2501.15564 (2025).
>
> [9] Soorchaei, Babak Ebrahimi, Arash Raftari, and Yaser Fallah. "Autonomous Systems and Intelligent Agents." Advances in Artificial Intelligence Applications in Industrial and Systems Engineering (2025): 19-42.
>
> [10] Muzahid, Abu Jafar Md, Xiaopeng Zhao,  and Zhenbo Wang. "Survey on human-vehicle interactions and AI  Collaboration for Optimal Decision-Making in Automated Driving." arXiv preprint arXiv:2412.08005 (2024).
>
> [11] Li, Yanwei, Chengyao Wang, and Jiaya Jia. "Llama-vid: An image is worth 2 tokens in large language models." European Conference on Computer Vision. Cham: Springer Nature Switzerland, 2024.
>
> [12] Qian, Rui, et al. "Streaming long video understanding with large language models." Advances in Neural Information Processing Systems 37 (2024): 119336-119360.
>
> [13] Hou, Jinghua, et al. "Query-based temporal fusion with explicit motion for 3d object detection." Advances in Neural Information Processing Systems 36 (2023): 75782-75797.
>
> [14] Zhang, Jiawei, et al. "Safeauto: Knowledge-enhanced safe autonomous driving with multimodal foundation models." arXiv preprint arXiv:2503.00211 (2025).

---

> ### Author Response · Authors · 2025-11-26
>
> Thank you again for the great efforts and valuable comments. We have carefully addressed the main concerns in detail. We hope you might find the response satisfactory. As the discussion phase is about to close, we are very much looking forward to hearing from you about any further feedback. We will be very happy to clarify any further concerns (if any).

---

### Author Response · Authors · 2025-12-02
**Global Rebuttal**

We sincerely thank the Area Chairs, Senior Area Chairs, and Program Chairs for their dedication and the additional effort invested in the review process during this challenging time for our community. To facilitate your assessment, we first summarize the **Key Common Concerns** and our corresponding response, followed by a **brief status overview** for each individual reviewer.

# Key Common Concerns
## 1. Concern 1: Inference Efficiency & Onboard Deployment (Raised by 2yL1, BksK, ScDp)

We address real-time deployment concerns by successfully deploying a Qwen2.5-VL-3B on a platform with comparable compute as the NVIDIA Thor X, where we achieve **~5Hz VLM inference** through **multi-token prediction (MTP) and quantization**. To further guarantee safety and mitigate latency jitter, we leverage an **asynchronous inference pipeline**, ensuring smooth real-time control. For more details, please refer to our reply to **Reviewer 2yL1 (Point 1)**.

## 2. Concern 2: Robustness to BEV Map Noise & Sensor Errors (Raised by 2yL1, BksK)

We clarified that nuPlan data is generated with real-world perception pipelines (instead of perfect ground truth). To further prove robustness, we conducted a **"Distance-Dependent Random Masking" stress test** (masking 20% of distant agents). BIRDriver maintains its great advantage over the baseline PLUTO even under noisy inputs (InterPlan score: ours-masked **54.11** vs PLUTO **48.92**).  For more details, please refer to our reply to **Reviewer BksK (Point 1)**.

## 3. Concern 3: Generalizability: New Planner & Loss Comparisons (Raised by ScDp, ctTg, 2yL1)

- New Planner: We replaced the backend planner PLUTO with **PlanTF**. BIRDriver(PlanTF) consistently outperforms vanilla PlanTF across all benchmarks, which prove that our framework is **model-agnostic**.

- Loss Function: We compared our Weighted SFT loss against the **PDCE loss**. Our Weighted SFT loss yields significantly lower regression errors.

For more details, please refer to our reply to **Reviewer ctTg (Point 1 & 4)**.

## 4. Concern 4: Dataset Choice (Why nuPlan instead of Waymo/nuScenes?) (Raised by 2yL1, ScDp, ctTg)

We clarified that **nuPlan offers Closed-Loop Reactive evaluation based on real-world logs**, which is superior to open-loop benchmarks (Waymo, nuScenes) that fail to reconstruct interactions. Our results on nuPlan demonstrate the actual driving capability, bridging the sim-to-real gap better than standard open-loop benchmarks.  For more details, please refer to our reply to **Reviewer 2yL1 (Point 2)** and **Reviewer ScDp (Point 2)**.

# Summary of Status by Reviewer

## Reviewer 2yL1
- **Addressed Common Concerns:** Efficiency, Robustness to Noise, Dataset Choice.
- **Specific Concern (Baselines & History):** The reviewer asked about Diffusion Planner comparisons and the 2s history limit.
- **Response:** We clarified that **Diffusion Planner is already included** as a primary baseline (Table 1), which BIRDriver significantly outperforms on the InterPlan benchmark. We also explained that the 2s history aligns with SOTA methods (PLUTO, PlanTF) for dynamic, reactive scenes.
## Reviewer BksK
- **Addressed Common Concerns:** Efficiency, BEV/Sensor Noise.
- **Specific Concern (Keypoint Count & Simple Scenarios):** The reviewer queried the rationale for using N=3 keypoints and noted a minor performance drop in simple scenarios.
- **Response:** We provided a kinematic justification (cubic spline requires 4 points: 1 fixed + 3 predicted) and experimental ablation showing N=3 is optimal. We explained that the minor Open-Loop (NR) drop is a trade-off for significant Closed-Loop (Reactive) gains and added noise injection during training to further mitigate this.
## Reviewer ScDp
- **Addressed Common Concerns:** Efficiency, Dataset Choice, Generalizability (New Planner).
- **Specific Concern (Novelty):** The reviewer questioned the novelty of using BEV inputs.
- **Response:** Our contribution is a **BEV-standardized hierarchical planner** relying solely on single-frame BEV maps, distinct from methods requiring auxiliary text or video describing the environment. We introduce a **sparse intention interface ($\le$3 keypoints)** that decouples strategic intent from trajectory refinement, unlike dense or language-based approaches (DriveVLM or Senna). In result, our system can leverage production-ready BEV streams to enable practical, **platform-agnostic deployment**.
## Reviewer ctTg
- **Addressed Common Concerns:** Generalizability (New Planner), Loss Function Comparison.
- **Specific Concern (VLM Necessity):** The reviewer asked if classical learning could replace the VLM for keypoints.
- **Response:** We argued that while classical methods are faster, they lack the **"common sense"** reasoning required for the long-tail/out-of-distribution scenarios that BIRDriver specifically targets.

---

### Meta-Review · Area_Chair_FnQQ · 2026-01-07

**Summary:**

This work proposes improvement over PLUTO hybrid planner by utilizing extra features generated by a VLM using BEV image. They fine tune the VLM to produce key points from BEV scene image using LoRA. These key points summarizes the trajectories of the agents in a compressed way. These are fed into the point encoding module of PLUTO planner to generate the trajectory. Their experiments show improvement on NuPlan and InterPlan over PLUTO.

Reviewers expressed the following concerns:
(1) The approach is only shown with PLUTO planner. It would have made the case stronger if the authors also showed benefit with at least one more planner with BEV based feature augmentation.
(2) A reviewer remarked that results are shown on nuPlan but not a reactive simulator; authors clarified that nuPlan closed-loop is reactive.
(3) Keypoint generation could use deterministic or classical learning for latency, and it is not clear what makes 3 the appropriate number of keypoints.
(4) Other loss functions could be compared.
(5) Lack of novelty of BEV maps or similar features as VLM inputs.
(6) A reviewer points out a fundamental concern: "Relying solely on BEV as input is overly idealistic and impractical. While the dataset currently in use contains BEV information, other datasets may not. More critically, ground-truth BEV information is not available during real-world driving. This would necessitate the model to predict the BEV, which in turn would introduce errors. Such a practical setting is far more complex than one with standard multi-view images, raising concerns about the model's generalizability and practical deployability."
(7) Lack of analysis of performance degradation in simple, routine scenarios, as well as sensor noise or occlusion scenarios.

**Reviewer Concerns:**

Authors sufficiently addressed (2) and (4) in their rebuttal, while leaving (3) for continued discussion; the authors highlight that they believe VLMs have some abilities to generate such keypoints, which reviewers contend could be handled classically. Authors addressed (1) by augmenting another baseline for evaluation. Addressing (5), authors clarify on the novelty of their method compared to other approaches that use BEV to enhance VLM planning.

The authors responded to the reviewers' concerns on (6) and (7); however, the BEV maps being generated offline for nuPlan would not satisfy the question of real-world applicability raised by the reviewer.

**Reviewer Scores:**

ctTg: increase (7), given the discussion that the closed-loop form of nuPlan was used for evaluation, and the addition of another baseline augmented by the proposed method.
ScDp: either increase to (5) or stay at (4), given the additional augmented baseline and possible continued discussion about the novelty of the proposed method.
BksK: stay at (4). Reviewers concerns are fundamental in nature, and not sufficiently addressed by the rebuttal.
2yL1: stay at (6). Concerns were adequately addressed, and no responses seemed to indicate misunderstandings from the reviewer, who already provided a high score.

---

### Decision · Program_Chairs · 2026-01-26

Accept (Poster)